# APG: Adaptive Parameter Generation Network for Click-Through Rate Prediction

**Bencheng Yan**[*], **Pengjie Wang**[*], **Kai Zhang, Feng Li, Hongbo Deng, Jian Xu, Bo Zheng** [†]
Alibaba Group
China
{bencheng.ybc,pengjie.wpj,victorlanger.zk,
adam.lf,dhb167148,xiyu.xj,bozheng}@alibaba-inc.com

## Abstract

In many web applications, deep learning-based CTR prediction models (deep CTR models for short) are widely adopted. Traditional deep CTR models learn patterns in a static manner, i.e., the network parameters are the same across all the instances. However, such a manner can hardly characterize each of the instances which may have different underlying distributions. It actually limits the representation power of deep CTR models, leading to sub-optimal results. In this paper, we propose an efficient, effective, and universal module, named as Adaptive Parameter Generation network (APG), which can dynamically generate parameters for deep CTR models on-the-fly based on different instances. Extensive experimental evaluation results show that APG can be applied to a variety of deep CTR models and significantly improve their performance. Meanwhile, APG can reduce the time cost by 38.7% and memory usage by 96.6% compared to a regular deep CTR model. We have deployed APG in the industrial sponsored search system and achieved 3% CTR gain and 1% RPM gain respectively.

## 1 Introduction

Recently, deep CTR models have achieved great success in various web applications such as recommender systems, web search, and online advertising [4, 9, 31, 14]. Formally, a regular deep CTR model can be expressed as $y_i = \mathcal{F}_\Theta(\boldsymbol{x}_i)$ where $\boldsymbol{x}_i, y_i$ are the input features and the predicted CTR of the instance $i$ respectively, $\Theta$ is the parameter, and $\mathcal{F}$ is usually implemented as a neural network.

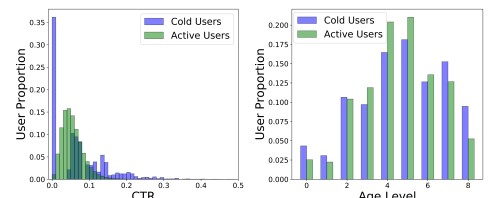

Figure 1: An example of feature distribution from different users (i.e., active vs. cold users). **Left:** the CTR distributions are varied from different user groups and a custom pattern should be considered. **Right:** A common pattern is welcomed to model the similar age distributions from different groups.

Improving the performance of deep CTR models has been a very hot topic in the research and industrial areas. Existing works can be broadly divided into two categories: (1) Focusing on $\boldsymbol{x}_i$, more and more elaborated information (e.g., user behavior features [37, 22], multimodal information [3, 11], knowledge graph [36, 30], etc) is introduced to enrich feature space (i.e., $\boldsymbol{x}_i$); (2) Focusing on $\mathcal{F}$, advanced architectures (including feature interaction modeling [4, 9, 31], automated architecture search [14, 26] and so on) are designed to improve the model performance.

---

[*]∗ These authors contributed equally to this work and are co-first authors.
[†]† Corresponding author

36th Conference on Neural Information Processing Systems (NeurIPS 2022).

However, few works focus on the improvement of the model parameters $\Theta$, especially for the weight matrix $\boldsymbol{W} \in \mathbb{R}^{N \times M}$ used in hidden layers of deep CTR models [3]. It is another orthogonal aspect for the performance improvement. Actually, most of the existing works simply adopt a static manner, i.e., all the instances share the same parameters $\boldsymbol{W}$. We argue that such a manner is sub-optimal for pattern learning and limits the representation power of deep CTR models. On the one hand, although the common patterns among instances can be captured by the shared parameters $\boldsymbol{W}$, it is not friendly to custom pattern modeling. Specifically, taking the industrial sponsored search system as an example, the feature distribution can be varied from different users (e.g., active vs. cold users), different categories (e.g., clothing vs. medicine), and so on (see Figure 1 (a) as an example). Simply applying the same parameters across all the instances can hardly capture the characteristic of each instance from different distributions. On the other hand, the learned common pattern may not be suitable for each of the instances. For example, the shared parameters tend to be dominated by high-frequency features and may give a misleading decision for the long-tailed instances. This leads us to the following question: *Do we really need the same and shared parameters for all instances?*

Ideally, besides modeling the common pattern, the parameters should be more adaptive and can be dynamically changed for different instances to capture custom patterns at the same time. Then, the representation power (or model capacity) can be enhanced by the dynamically changed parameter space. To achieve this goal, we design a new paradigm for CTR prediction. The key insight is to propose an Adaptive Parameter Generation network (APG) to dynamically generate parameters depending on different instances. Firstly, we propose a basic version (Section 3.1) of APG which can be expressed as $y_i = \mathcal{F}_{\mathcal{G}(\boldsymbol{z}_i)}(\boldsymbol{x}_i)$ where $\mathcal{G}$ refers a neural network (e.g., MLP) and generates the adaptive parameters $\boldsymbol{W}_i \in \mathbb{R}^{N \times M}$ by the input-aware condition $\boldsymbol{z}_i \in \mathbb{R}^D$. However, the basic model suffers from two problems: (1) inefficient in time and memory. Directly generating the weight matrix $\boldsymbol{W}_i$ of a deep CTR model needs $\mathcal{O}(NMD)$ cost in computation and memory storage, which is $D$ times the cost in a regular deep CTR model and is costly especially for a web-scale application where $N, M$ are usually set as a large value (e.g., $N=M=1{,}000$). The empirical results (Section 4.4) also show the basic model needs an extra $111\times$ training time and $31\times$ memory usage. (2) sub-effective in pattern learning. The parameter generation process is totally dependent on the condition $\boldsymbol{z}_i$, which may only capture the custom patterns and ignore the common patterns which contribute to understand users' behaviors (see Figure 1 (b)), leading to sub-effective pattern learning.

Then, we extend the above basic model to an efficient and effective version of APG: **(1) For the efficiency,** motivated by the low-rank methods [16, 1] which show that the weight matrix resides on a low intrinsic dimension, we parameterize the target weight matrix $\boldsymbol{W}_i$ as the production of three low-rank matrices $\boldsymbol{U}_i \boldsymbol{S}_i \boldsymbol{V}_i$ where $\boldsymbol{U}_i \in \mathbb{R}^{N \times K}, \boldsymbol{V}_i \in \mathbb{R}^{K \times M}, \boldsymbol{S}_i \in \mathbb{R}^{K \times K}$ and $K \ll min(M, N)$. In addition, the decomposed feed-forwarding is proposed to avoid the heavy computation of the weight matrix $\boldsymbol{W}_i$ reconstruction. Then we further take the center matrix $\boldsymbol{S}_i$ as the specific parameters which are dynamically generated to capture the custom patterns and the rest two matrices $\boldsymbol{U}, \boldsymbol{V}$ as the shared parameters which are randomly initialized and shared across instances to capture common patterns. In this way, the complexity of generating the specific parameters can be easily controlled by setting a small $K$. As a result, APG achieves $\mathcal{O}(KKD+NK+MK+KK)$ time complexity and $\mathcal{O}(KKD+NK+MK)$ memory complexity compared to that both have $\mathcal{O}(NM)$ in a regular deep CTR model. We empirically find APG can speed up the training time by 38.7% and reduce memory usage by 96.6% relative to a regular deep CTR model (Section 4.4). **(2) For the effectiveness,** apart from the natural effectiveness of APG in custom pattern learning, to model the common pattern, the shared weights $\boldsymbol{U}$ and $\boldsymbol{V}$ are considered in the adaptive parameter generation. Then we further extend $\boldsymbol{U}$ and $\boldsymbol{V}$ to an over parameterization version which enriches the model capacity without any additional memory and time cost during inference. Besides, we also find there exists inherent similarity between different generated $\boldsymbol{S}_i$ which implicitly model the common information (Section 4.6).

In summary, the contributions of this paper are presented as follows: (1) We propose a new learning paradigm in deep CTR models where the model parameters are input-aware and dynamically generated to boost the representation power. It is orthogonal to many prior methods and is a universal module that can be easily applied in most existing deep CTR models. (2) We present APG which generates the adaptive parameters in an efficient and effective way, and theoretically analyze the

---

[3]For simplicity, in this paper, we mainly discuss the weight matrix $W$. Our method can also be easily applied to the parameters of other modules (e.g., transformer, attention network, etc) in deep CTR models since most of them can be regarded as a variety of MLP with a set of weight matrices [35].

computation and memory complexity. (3) Extensive experimental evaluation results demonstrate that the proposed method is a universal module and can improve the performance of most of the existing deep CTR models. we also provide a systematic evaluation of APG in terms of training time and memory consumption. Finally, we have developed APG in the industrial sponsored search system and achieved 3% CTR gain and 1% RPM gain respectively.

## 2  Related Work

**Deep CTR Models.** A traditional CTR prediction method usually adopts a deep neural network to capture the complex relations between users and items. Most of them focus on (1) introducing abundant useful informations [37, 22, 3, 11, 36, 30] to improve the model understanding, (2) designing advanced architecture [4, 9, 31, 14, 26] to achieve better performance. All of them adopt a static parameters manner which limits the model capacity of these methods, leading to sub-optimal performance

**Coarse-grained Parameter Allocating.** There are some research areas that bring a coarse-grained parameter allocating strategy that may be related to our goals, including multi-domain learning [25, 15] and multi-task learning [19, 21]. Both of them maintain and allocate different parameters to different domains or tasks manually. However, such a coarse-grained parameter allocating can hardly be extended to a fine-grained (e.g., user, item, or instances sensitive) manner. It not only costs too much memory to maintain and store a large number of parameters when considering the fine-grained modeling but also lacks flexibility and generalization since the parameter allocation is manually pre-defined. We also conduct experiments to compare this kind of methods (Section 4.5).

**Dynamic Deep Neural Networks.** Our method is related to recent works of dynamic neural networks used in computer vision and natural language processing, in which the model parameters [13, 32, 7, 23, 10, 20, 29] and architecture [17, 34] can be dynamically changed. In our paper, we take the first step to bring the idea about dynamic networks into deep CTR models and develop it in real applications where the efficiency in computation and memory is extremely needed and the common and custom pattern learning are also required.

## 3  Method

In this paper, we denote scalars, vectors and tensors with lower-case (or upper-case), bold lower-case, and bold upper-case letters, e.g., $n$ (or $N$), $\boldsymbol{x}$, $\boldsymbol{X}$, respectively.

### 3.1  Basic Model

In this section, we introduce the basic version of APG. Generally speaking, the basic idea of APG is to dynamically generate parameters $\boldsymbol{W}_i$ by different condition $\boldsymbol{z}_i$, i.e., $\boldsymbol{W}_i = \mathcal{G}(\boldsymbol{z}_i)$ where $\mathcal{G}$ refers to the adaptive parameter generation network. Then the generated parameters are applied to the deep CTR models, i.e., $y_i = \mathcal{F}_{\mathcal{G}(\boldsymbol{z}_i)}(\boldsymbol{x}_i)$. Next, we present (1) how to design the condition $\boldsymbol{z}_i$ for different instance $i$ and (2) how to implement $\mathcal{G}$.

#### 3.1.1  Condition Design

In this paper, we propose three kinds of strategies (including group-wise, mix-wise, and self-wise) to design different kinds of $\boldsymbol{z}_i$.

**Group-wise.** The group-wise strategy tries to generate different parameters depending on different instance groups. The purpose is that, sometimes, the instances can be divided into different groups [2] and instances in the same group may have similar patterns. Thus $\boldsymbol{z}_i$ is the representation identifying different groups. An example can be found in Appendix A.1.

**Mix-wise.** To further enrich the expression power and flexibility of the adaptive parameters, a mix-wise strategy is designed to take multiple conditions into consideration. Specifically, given $k$ condition embeddings $\{\boldsymbol{z}_i^j \in \mathbb{R}^d | j \in \{0, 1, .., k-1\}\}$ of the instance $i$, we propose two aggregation policies to consider different conditions at the same time: (1) *Input Aggregation.* This policy firstly aggregates different condition embeddings and then feeds the aggregated embeddings into APG to obtain the mix-wise based parameters. The aggregation functions can be (but are not limited to):

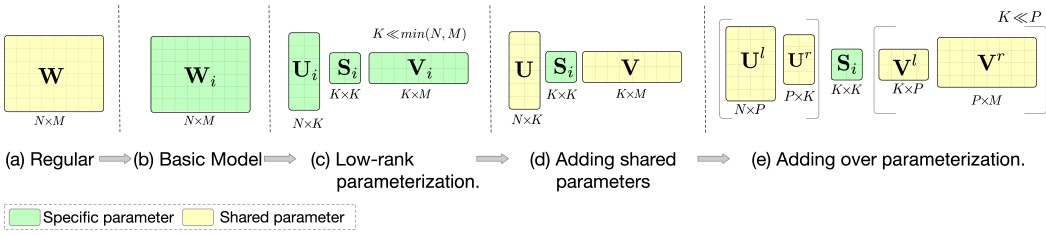

(a) Regular ⟹ (b) Basic Model ⟹ (c) Low-rank parameterization. ⟹ (d) Adding shared parameters ⟹ (e) Adding over parameterization.

☐ Specific parameter  ☐ Shared parameter

Figure 2: The comparison of different versions of APG.

Concatenation, Mean, and Attention. (2) *Output Aggregation* This policy firstly feeds different $z_i^j$ into different APG and obtains the corresponding parameters $\boldsymbol{W}_i^j$ respectively. Then the aggregation is applied to these adaptive parameters $\boldsymbol{W}_i^j$. Similarly, the aggregation function includes Concatenation, Mean, and Attention. Examples can be found in Appendix A.2.

**Self-wise.** The above two strategies need additional prior knowledge to generate parameters. Self-wise strategy tries to use a simple and easily obtained knowledge (i.e., self-knowledge) to guide parameters generation, i.e., the network parameters are generated by their own input. For example, for the 0-th hidden layer of a deep CTR model, we can set $\boldsymbol{z}_i = \boldsymbol{x}_i$. For $l$-th hidden layer, $\boldsymbol{z}_i = \boldsymbol{h}^{l-1}$ where $\boldsymbol{h}^{l-1}$ is the input of the $l$-th hidden layer.

### 3.1.2 Parameters Generation

After obtaining the conditions, we adopt a multilayer perceptron[4] to generate parameters depending on these conditions (see Figure 2 (b)), i.e.,

$$\boldsymbol{W}_i = reshape(MLP(\boldsymbol{z}_i)) \tag{1}$$

where $\boldsymbol{W}_i \in \mathbb{R}^{N \times M}$, $\boldsymbol{z}_i \in \mathbb{R}^D$, and the operation $reshape$ refers to reshaping the vectors produced by the MLP into a matrix form. Then a deep CTR model with APG can be expressed as:

$$y_i = \sigma(\boldsymbol{W}_i \boldsymbol{x}_i) \tag{2}$$

where $\sigma$ is the activation function. Here we take a deep CTR model with an MLP layer as an example, and other deep CTR models can also be easily extended since most of them can be regarded as a variety of MLP with a set of weight matrices [35].

### 3.2 Effective and Efficient Adaptive Parameter Generation Network

As introduced in Section 1, the above basic model has two problems: (1) inefficient in time and memory and (2) sub-effective in pattern learning. To address the above two problems, we propose some extensions including low-rank parameterization, decomposed feed-forwarding, parameter sharing, and over parameterization to parameterize the weight matrix in an efficient and effective way.

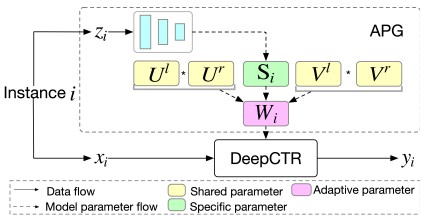

Figure 3: The framework of APG. For simplicity, the process of decomposed feed-forwarding is omitted in this figure.

**Low-rank parameterization.** Inspired by the recent success of the low-rank methods [16, 1] which have demonstrated that strong performance can be achieved by optimizing a task in a low-rank subspace, we hypothesize that the adaptive parameters also have a low "intrinsic rank". To this end, we propose to parameterize the weight matrix $\boldsymbol{W}_i \in \mathbb{R}^{N \times M}$ as a low-rank matrix, which is the product of three sub-matrices $\boldsymbol{U}_i \in \mathbb{R}^{N \times K}, \boldsymbol{S}_i \in \mathbb{R}^{K \times K}, \boldsymbol{V}_i \in \mathbb{R}^{K \times M}$ and the rank $K \ll min(N, M)$ (see Figure 2 (c)). Formally, the weight generation process can be expressed as:

$$\boldsymbol{U}_i, \boldsymbol{S}_i, \boldsymbol{V}_i = reshape(MLP(\boldsymbol{z}_i)) \tag{3}$$

---

[4]In this paper, we take MLP as an example, and other implementations of $\mathcal{G}$ can also be considered.

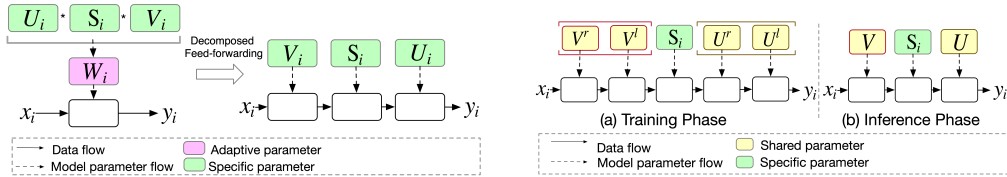

Figure 4: An example of the decomposed feed-forwarding.

Figure 5: An example of over parameterization.

Intuitively, we can set a small value of $K$ to control the time and memory cost. Meanwhile, ignoring the substantial storage and computation cost, $K$ can also be set to a higher value to enlarge the adaptive parameter space (see detailed discussion in Appendix E). Overall, through the low-rank parameterization, we can significantly reduce dimensionality in the parameter space, i.e., $K \ll min(N, M)$, to enable a more compact model. As a result, the computation complexity of the specific parameter generation process can be reduced to $\mathcal{O}((NK+MK+KK)D)$ (see Table 1). The memory cost is also reduced to $\mathcal{O}((NK+MK+KK)D)$ (see Table 1).

**Decomposed Feed-forwarding.** After low-rank parameterization, the Eq 2 can be written as

$$y_i = \sigma(\boldsymbol{W}_i \boldsymbol{x}_i) = \sigma((\boldsymbol{U}_i \boldsymbol{S}_i \boldsymbol{V}_i) x_i) = \sigma(\boldsymbol{U}_i(\boldsymbol{S}_i(\boldsymbol{V}_i \boldsymbol{x}_i))) \tag{4}$$

Here instead of directly reconstructing the weight matrix $\boldsymbol{W}_i$ by the sub-matrix production, we design a decomposed feed-forwarding and apply $\boldsymbol{x}_i$ to each sub-matrix sequentially (see Figure 4). Such a design helps us avoid the heavy computation of the sub-matrix production which costs $\mathcal{O}(NKK+NMK)$. Actually, this design benefits from the low-rank parameterization which naturally supports the decomposed feed-forwarding. Since the computation complexity in Eq 4 is $\mathcal{O}(NK+KK+MK)$, the total time cost of a deep CTR model with APG per layer is reduced to $\mathcal{O}((NK+MK+KK)(D+1))$ (see Table 1).

**Parameter sharing.** In this section, we present our common pattern modeling strategy. Thanks to decomposing the weight matrix into $\boldsymbol{U}_i, \boldsymbol{S}_i$, and $\boldsymbol{V}_i$, it allows us to be more flexible. Consequently, we divide these three matrices into (1) *specific parameters* that capture custom patterns from different instances; (2) *shared parameters* that are shared across instances to characterize the common patterns. Specifically, we define $\boldsymbol{S}_i$ as the specific parameters since the matrix scale is totally controlled by $K$ and is more efficient for the generation process. $\boldsymbol{U}$ and $\boldsymbol{V}$ are regarded as the shared parameters (see Figure 2 (d)). Then we can rewrite Eq 3 and 4 as:

$$\boldsymbol{S}_i = reshape(MLP(\boldsymbol{z}_i)) \tag{5}$$

$$y_i = \sigma(\boldsymbol{U}(\boldsymbol{S}_i(\boldsymbol{V}\boldsymbol{x}_i))) \tag{6}$$

Furthermore, such a design also contributes the efficiency. Since the size of the generated specific parameter is reduced to $K \times K$, the computation complexity of the specific parameter generation can be further reduced to $\mathcal{O}(KKD)$ in Eq 5, and the total time cost of a deep CTR model with APG is reduced to $\mathcal{O}(KKD+NK+MK+KK)$ (see Table 1). Meanwhile the memory cost is reduced to $\mathcal{O}(KKD+NK+MK)$ where *NK* and *MK* refers to the storage of the shared parameters $\boldsymbol{U}$ and $\boldsymbol{V}$ respectively (see Table 1).

**Over Parameterization** Compared with the shared weight matrix $\boldsymbol{W} \in \mathbb{R}^{N \times M}$ in a regular deep CTR model, $\boldsymbol{U}$ and $\boldsymbol{V}$ in APG can be hardly scaled to large matrices due to the efficiency constraint with $K \ll min(N, M)$, leading to a possible performance drop. To address this problem and further enlarge the model capacity, we follow the idea about the over parameterization [1, 5] to enrich the number of the shared parameters (see Figure 2 (e)). Specifically, we replace shared parameters in Eq 6 with two large matrices, i.e.,

$$\boldsymbol{U} = \boldsymbol{U}^l \boldsymbol{U}^r, \ \boldsymbol{V} = \boldsymbol{V}^l \boldsymbol{V}^r \tag{7}$$

where $\boldsymbol{U}^l \in \mathbb{R}^{N \times P}, \boldsymbol{U}^r \in \mathbb{R}^{P \times K}, \boldsymbol{V}^l \in \mathbb{R}^{K \times P}, \boldsymbol{V}^r \in \mathbb{R}^{P \times M}$ and $P \gg K$. Although $\boldsymbol{U}$ (or $\boldsymbol{V}$) is exactly equal to $\boldsymbol{U}^l \boldsymbol{U}^r$ (or $\boldsymbol{V}^l \boldsymbol{V}^r$) at the mathematical perspective, replacing $\boldsymbol{V}$ with $\boldsymbol{V}^l \boldsymbol{V}^r$ can contribute into two folds: (1) Since $P \gg K$, more shared parameters are introduced to enlarge the model capacity [5, 6]; (2) The form as the multiple matrix production can result in an implicit regularization and thus enhance generalization [1].

*No Additional Inference Latency and Memory Cost.* During the training phase, we can set $P$ to a large value to enlarge the model representation power. Remarkably, when $P > max(N, M)$, the shared

Table 1: The computation and memory complexity per layer of different versions of APG. SPG refers to the time or memory cost in the process of specific parameter generation. R-Wi is the time cost to reconstruct $W_i$. FF refers to the time cost in the feed-forwarding process of a regular deep CTR model. SPS refers to the memory cost to store the shared parameters. The total computation cost is the sum of SPG, R-Wi, and FF. The total memory cost is the sum of SPG and SPS. Since over parameterization dose not introduce any additional time cost, it is not depicted here.

| Versions | Computation complexity per layer | | | | Memory complexity per layer | | |
|---|---|---|---|---|---|---|---|
| | SPG | R-Wi | FF | Total Cost | SPG | SPS | Total Cost |
| $Wx_i$ | - | - | $\mathcal{O}(NM)$ | $\mathcal{O}(NM)$ | - | $\mathcal{O}(NM)$ | $\mathcal{O}(NM)$ |
| $W_ix_i$ | $\mathcal{O}(NMD)$ | - | $\mathcal{O}(NM)$ | $\mathcal{O}(NMD+NM)$ | $\mathcal{O}(NMD)$ | - | $\mathcal{O}(NMD)$ |
| $(U_iS_iV_i)x_i$ | $\mathcal{O}((NK+MK+KK)D)$ | $\mathcal{O}(NKK+NMK)$ | $\mathcal{O}(NM)$ | $\mathcal{O}((NK+MK+KK)D+NKK+NMK+NM)$ | $\mathcal{O}((NK+MK+KK)D)$ | - | $\mathcal{O}((NK+MK+KK)D)$ |
| $U_i(S_i(V_ix_i))$ | $\mathcal{O}((NK+MK+KK)D)$ | - | $\mathcal{O}(NK+MK+KK)$ | $\mathcal{O}((NK+MK+KK)(D+1))$ | $\mathcal{O}((NK+MK+KK)D)$ | - | $\mathcal{O}((NK+MK+KK)D)$ |
| $U(S_i(Vx_i))$ | $\mathcal{O}(KKD)$ | - | $\mathcal{O}(NK+MK+KK)$ | $\mathcal{O}(KKD+NK+MK+KK)$ | $\mathcal{O}(KKD)$ | $\mathcal{O}(NK+MK)$ | $\mathcal{O}(KKD+NK+MK)$ |

parameter space can be larger than that in a regular deep CTR model. During the inference phase, we explicitly pre-compute and store $V, U$, and use these two matrices for inference (see Figure 5). Critically, this guarantees that we can introduce abundantly shared parameters without any additional latency and memory cost during the inference phase.

### 3.3 Complexity
In this section, we detailedly analyze the proposed model complexity including the memory and computation complexity during the inference phase. For analysis, the parameters generation network is implemented as a single perceptron layer, and the per layer costs in a regular deep CTR model and an adaptive deep CTR model are compared. Summarization can be found in Table 1.

**Memory Complexity.** For a regular deep CTR model, the memory cost is $\mathcal{O}(NM)$ per layer, i.e., storing the shared weight matrix. For APG, the memory cost comes from two parts: (1) The memory cost of generating $S_i$ in Eq 5 is $\mathcal{O}(KKD)$; (2) The shared parameters $U, V$ cost $\mathcal{O}((N + M)K)$ during the inference phase. Then the total memory complexity of APG is $\mathcal{O}(KKD+NK+MK)$ per layer.

**Computation Complexity.** A regular deep CTR model needs $\mathcal{O}(NM)$ per layer for the feed-forward computation. APG needs $\mathcal{O}(KKD)$ in Eq 5 to calculate the specific parameters. Meanwhile, the feed-forwarding computation of a deep CTR model is $\mathcal{O}(NK+KK+MK)$ by the decomposed feed-forwarding in Eq 8. In total, the computation complexity is $\mathcal{O}(KKD+NK+KK+MK)$.

In summary, APG has $\mathcal{O}(KKD+NK+MK)$ in memory cost and $\mathcal{O}(KKD+NK+KK+MK)$ in computation cost. Since $K \ll min(N, M)$ and $D$ is usually set smaller than $N, M$, the memory and computation cost of APG can even be much smaller than that (i.e., $\mathcal{O}(NM)$) in a regular deep CTR model. The experimental results in Section 4.4 also show the efficiency of APG.

## 4 Experiments
### 4.1 Experimental Settings

The detailed settings including datasets, baselines, and training details are presented in Appendix B.

**Datasets.** Four real-world datasets are used including **Amazon**, **MovieLens**, **IAAC**, and **IndusData**. The first three are public datasets and the last is an industrial dataset.

**Baselines.** Here, we compare our method with two kinds of methods (1) *Existing CTR prediction methods* include WDL[4], PNN[24], FIBINET[12], DIFM[18], DeepFM[9], DCN[31], and AutoInt[27]; (2) *Coarse-grained parameter allocating methods.* multi-task learning: MMoE [19] and multi-domain learning: Star [25].

**Training Details.** See Appendix B.3 for the detailed introduction.

### 4.2 Performance Evaluation with Existing Deep CTR Models

**Results on public datasets.** We first apply APG on CTR prediction tasks on public datasets. APG is a universal module and can be applied to most of the existing deep CTR models. Hence, to evaluate the performance of APG, we apply APG to various existing deep CTR models, and report the results of the original model (denoted as Base) and the model with APG. Here, AUC (%) [8] score is reported as the metric [5]. The results are shown in Table 2. We can find that with the help of APG, all of

---

[5]Note 0.1% absolute AUC gain is regarded as significant for the CTR task [37, 28, 4].

Table 2: The AUC (%) results of Click-Through Rate (CTR) prediction on different datasets. Note Base refers to the original results of the corresponding methods and Base+APG refers to the results with the help of APG. Ave is the average results across all cases. $\Delta$ refers the improvement of Base+APG compared to Base.

| Data | Method | WDL | PNN | FIBINET | DIFM | DeepFM | DCN | AutoInt | Ave |
|------|--------|-----|-----|---------|------|--------|-----|---------|-----|
| MovieLens | Base | 79.21 | 79.5 | 79.78 | 79.84 | 79.3 | 79.29 | 79.36 | 79.46 |
| | Base+APG | **79.73** | **79.67** | **79.82** | **79.94** | **79.60** | **79.62** | **79.64** | **79.70** |
| | $\Delta$ | +0.39 | +0.17 | +0.04 | +0.10 | +0.30 | +0.33 | +0.28 | +0.24 |
| Amazon | Base | 69.15 | 69.16 | 68.88 | 69.17 | 69.1 | 68.98 | 68.96 | 69.06 |
| | Base+APG | **69.43** | **69.37** | **69.19** | **69.23** | **69.43** | **69.42** | **69.38** | **69.34** |
| | $\Delta$ | +0.22 | +0.21 | +0.31 | +0.06 | +0.33 | +0.44 | +0.42 | 0.28 |
| IAAC | Base | 65.17 | 65.3 | 65.15 | 65.76 | 65.64 | 64.78 | 64.99 | 65.26 |
| | Base+APG | **65.94** | **65.87** | **66.15** | **66.42** | **66.17** | **66.39** | **66.21** | **66.15** |
| | $\Delta$ | +0.77 | +0.57 | +1.0 | +0.66 | +0.53 | +1.61 | +1.22 | +0.91 |

Table 3: The settings of different APG versions.

| | Version | Annotation |
|------|---------|------------|
| Base | $Wx_i$ | WDL [4] as the baseline and the backbone |
| v1 | $W_i x_i$ | Basic model (Section 3.1) |
| v2 | $(U_i S_i V_i)x_i$ | + Low-rank parameterization |
| v3 | $U_i(S_i(V_i x_i))$ | + Decomposed feed-forwarding |
| v4 | $U(S_i(V x_i))$ | + Parameter sharing |
| v5 | $(U^t U^\tau)(S_i((V^t V^\tau)x_i))$ | + Over parameterization |

Table 4: The AUC results of the evaluation of different versions of APG. $\Delta$ refers the difference relative to Base.

| | MovieLens | Amazon | IAAC | Ave(AUC) | Ave($\Delta$) |
|------|-----------|--------|------|----------|---------------|
| Base | 79.21 | 69.15 | 65.17 | 71.18 | — |
| v1 | 79.51 | 69.33 | 65.52 | 71.45 | +0.27 |
| v2 | 79.49 | 69.24 | 65.61 | 71.45 | +0.27 |
| v4 | 79.61 | 69.36 | 65.78 | 71.58 | +0.40 |
| v5 | 79.73 | 69.43 | 65.94 | 71.70 | +0.52 |

the methods achieve a significant improvement on all datasets. For example, the gains of DCN is $0.33\% \sim 1.61\%$ (other methods also can obtain similar improvement). It demonstrates that (1) giving adaptive parameters for models can enrich the parameter space and learn more useful patterns for different instances; (2) the proposed APG is a universal framework that can boost the performance of many other methods. Such nice property encourages APG to be applied to various scenarios and various methods.

**Results on industrial application.** We also develop APG in the industrial sponsored search system, and achieve 0.2% AUC gain on the industrial dataset, 3% CTR gain and 1% RPM (Revenue Per Mile) gain during online A/B test. Detailed analysis are presented in Appendix F.

### 4.3 Effectiveness Evaluation

In this section, we implement various versions of APG (see Table 3) and conduct experiments to detailedly evaluate the effectiveness of APG. The AUC results are reported in Table 4. Note since the design of decomposed feed-forwarding does not influence the AUC performance of APG, we do not compare version v3 in this section.

**The impact of the basic model.** Compared with the base, v1 achieves significant improvements on AUC results overall datasets. It demonstrates the effectiveness to introduce specific parameters to give a custom understanding of different instances.

**The impact of the low-rank parameterization.** Since the parameter matrix resides on a low intrinsic dimension [16, 1], it encourages a low-performance drop when adopting a low-rank based method to reduce the computation and memory cost. Considering the effectiveness, v1 and v2 do not provide much performance difference. Both of them achieve high performance and perform much better than Base. More importantly, by low-rank parameterization, we can generate adaptive parameters in a more efficient way (see Section 4.4 for details).

**The impact of the parameter sharing.** In APG, we introduce the shared parameters to characterize common patterns. Comparing the performance of versions v2, and v4 can further improve the performance by introducing the shared parameters, demonstrating the effectiveness of APG in common pattern modeling. Furthermore, sharing parameter also contributes to the efficiency, due to fewer specific parameters generated (see Section 4.4).

Table 5: The inference time and memory cost for different versions of APG. $\Delta$ is the relative difference with respect to Base.

| Method | [393,64,32,16] | | [393,128,64,32] | | [393,256,128,64] | | [393,512,256,128] | | [393,1024,512,256] | |
|---|---|---|---|---|---|---|---|---|---|---|
| | Cost | $\Delta$ | Cost | $\Delta$ | Cost | $\Delta$ | Cost | $\Delta$ | Cost | $\Delta$ |
| **Time (s)** | | | | | | | | | | |
| Base | 4.52 | — | 5.27 | — | 5.89 | — | 7.12 | — | 13.04 | — |
| v1 | 16.52 | 265.5% | 50.31 | 854.6% | 126.27 | 2043.8% | 420.33 | 5803.5% | 1462.78 | 11117.6% |
| v2 | 13.83 | 206.0% | 39.53 | 682.8% | 90.43 | 1435.3% | 363.07 | 4999.3% | 1065.35 | 8069.9% |
| v3 | 6.31 | 39.6% | 5.42 | 7.3% | 5.71 | -3.1% | 6.31 | -11.4% | 11.23 | -13.9% |
| v4 | 4.49 | -0.7% | 4.78 | -5.3% | 5.46 | -7.3% | 5.89 | -17.3% | 7.99 | -38.7% |
| **Memory (M)** | | | | | | | | | | |
| Base | 0.50 | — | 0.89 | — | 1.93 | — | 4.61 | — | 12.91 | — |
| v1 | 11.21 | 2160.1% | 24.20 | 2613.0% | 56.32 | 2818.1% | 144.84 | 3041.9% | 419.11 | 3146.4% |
| v2 | 0.74 | 50.0% | 0.91 | 1.7% | 1.21 | -37.3% | 1.98 | -57.0% | 3.22 | -75.1% |
| v4 | 0.27 | -45.0% | 0.29 | -68.0% | 0.31 | -84.1% | 0.35 | -92.4% | 0.44 | -96.6% |

**The impact of the over parameterization.** Comparing the performance with or without over parameterization (i.e., v4 vs. v5), it shows that v5 performs better than v4 in all cases, which indicates adding more shared parameters can enrich the model capacity and lead to better performance.

In addition, we also give a detailed analysis about the impact of the condition design in Appendix D and C, and the impact of the hyper-parameters in Appendix E. Furthermore, the influence to different frequency instances are also discussed in Appendix F

## 4.4 Efficiency Evaluation

In this section, we evaluate the time and memory efficiency of our proposed method. To this end, we train different versions (see Table 3) of APG on the dataset IAAC and analyze the influence of each extension introduced in Section 3.2. Since over parameterization does not introduce any cost, it is not considered here. The decomposed feed-forwarding does not bring extra memory cost, it is not discussed in the memory usage. For all versions, we set $K = 4$ and the backbone (also the base) is WDL with 3 hidden layers. We gradually increase the number of hidden units from [64,32,16] to [1024,512,256] with a fixed input_shape=393 to evaluate the memory and time cost in different model scales. In Table 5, we report the inference time per epoch, memory usage[6] of each version.

For the basic model (v1), although it achieves high performance (see Table 4), it is time expensive (265.5% $\sim$ 11117.6% relative to Base) and memory costly (2160.1% $\sim$ 3146.6% relative to Base). Such an inefficient model can hardly be accepted in web-scale applications.

When introducing the low-rank parameterization (see v2 in Table 5), compared to the basic model, the inefficiency problem in v2 is addressed across all cases. Moreover, compared with Base, v1 can reduce memory usage (e.g., -75.1% in the large scale model). Note when we set $N, M$ to a small value (e.g., [64,32,16] or [128,64,32]), the memory cost, theoretically in $\mathcal{O}((NK+MK)D)$, is more sensitive with $K$ and $D$, leading to a little increasement. For the time efficiency, the contribution of the low-rank parameterization can be summarized as follows: (1) Although v2 still needs a high time requirement due to the weight matrix $W_i$ reconstruction, compared with v1, the overall time cost of v2 is decreased. (2) The low-rank parameterization naturally contributes to the decomposed feed-forwarding which plays a key role in efficient learning.

For version v3 which adopts decomposed feed-forwarding, it is free from the high computation of reconstructing the weight matrix $W_i$ and achieves great improvement by -13.9% in time cost when considering a large scale model. In addition, since v3 does not introduce any extra memory cost, it has the same memory usage as v2. Note in the small model, the computation is far from the GPU bottleneck and time cost is insensitive with GFlops, leading to less improvement.

---

[6]Since in this paper we mainly focus on the improvement of the hidden layers, we only count the memory cost of these hidden layers and the other parts (e.g., embedding layers) are not included here.

Table 6: The comparison with coarse-grained parameter allocating methods. Mem refers to the memory cost (M) of the hidden layers.

| | MovieLens | | | Amazon | | | IAAC | | |
|---|---|---|---|---|---|---|---|---|---|
| | Setting | AUC | Mem | Setting | AUC | Mem | Setting | AUC | Mem |
| Base | — | 79.21 | 1.29 | – | 69.15 | 1.51 | — | 65.17 | 1.93 |
| Base+MMoE$_{fg}$ | movie | 79.31 | 123.42 | item | 69.17 | 315.93 | item | 65.22 | 332.97 |
| Base+MMoE$_{cg}$ | movie gender | 79.28 | 32.91 | item category | 69.14 | 5.78 | item brand | 65.20 | 70.53 |
| Base+Star$_{fg}$ | movie | 79.28 | 3784.23 | item | 69.20 | 11034.02 | item | 65.36 | 13753.71 |
| Base+Star$_{cg}$ | movie gender | 79.25 | 311.56 | item category | 69.20 | 42.84 | item brand | 65.28 | 2866.88 |
| Base+APG$_{fg}$ | movie | **79.58** | 0.24 | item | **69.35** | 0.26 | item | **65.76** | 0.38 |
| Base+APG$_{cg}$ | movie gender | 79.46 | 0.24 | item category | 69.30 | 0.26 | item brand | 65.59 | 0.38 |

When introducing the sharing parameters, v4 only needs to generate a small weight matrix, where time and memory cost (i.e., $\mathcal{O}(KKD)$) in the generation process are free from the scales of a model. Thus, the memory requirement of v4 is best among all cases, reducing memory by -45% to -96.6% relative to Base. v4 also speeds up inference time substantially, by -0.7% to -38.7% relative to Base. It also supports the theoretical complexity analysis in Section 3.3.

Overall, such nice properties in terms of high performance, efficient memory usage, and low time requirement of our proposed model are welcomed for web-scale applications.

## 4.5 Comparison with coarse-grained parameter allocating methods

In this section, we compare the performance with coarse-grained parameter allocating methods (CGPMs), including Star and MMoE. Specifically, we conduct two settings (see Table 6), i.e., coarse-grained (cg) and fine-grained (fg), for each method. For the former, methods adopt coarse-grained parameter modeling strategies, e.g., developing different parameters for different movie genders in MovieLens. For the latter, methods adopt fine-grained parameter modeling strategies, e.g., developing different parameters for different movies in MovieLens. The results are presented in Table 6. We can find, compared with Star and MMoE, APG shows superiority in efficiency and effectiveness. The reasons are that (1) the specific parameters of APG are dynamically generated on-the-fly, without any need to store these specific parameters, and the architecture of APG is also designed efficiently to further save memory. For Star and MMoE, they have to store these specific parameters due to the manually allocating strategies used in CGPMs. Even worse, when comes to a fine-grained setting, more parameters are maintained. (2) APG has better generalization than Star and MMoE, leading to better performance. Actually, APG adopts parameter generation manner which provides a potential opportunity for generalization, and it has the ability to imply the inherent similarity between different specific parameters (see Section 4.6 for additional experiments). Especially for the fine-grained setting where there may be a lack of enough instances to train the specific parameters, the inherent connections among different specific parameters are helpful for parameter learning.

## 4.6 Visualization

In this section, we visualize the generated specific parameters. Specifically, we take the group-wise strategy on Amazon as an example. The item category is set as the condition, and there is a total of 10 different categories. Then we plot the generated specific parameters (i.e., $S_i$) into a 2-D space by PCA [33]. The visualization is presented in Figure 6. Each point refers to the specific parameters generated for one specific category. Interestingly, the observed groupings (i.e, in the same dashed circle) correspond to similar categories. This shows that the learned specific parameters by APG are meaningful and can capture the relations among different specific parameters implicitly.

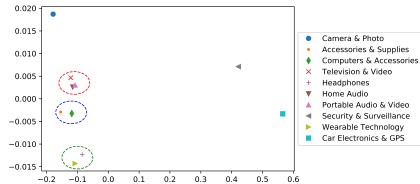

Figure 6: The visualization of the generated specific parameters.

## 5 Conclusion

In this paper, we propose an efficient, effective, and universal module to adaptively generate parameters for different instances. In this way, the model can carefully characterize the patterns for different instances by adopting different parameters. Experimental results show that with the help of APG, all of the existing deep CTR models can make great improvements, which also encourages a wide application for APG. Furthermore, the effectiveness and efficiency of APG are also analyzed in detail. Currently, APG requires users to set some hyper-parameters, e.g., condition strategies, $K$, $P$, and etc. In the future, we will attempt to automatically implement APG with different settings for different situations.

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
