# A supplementary material - APG: Adaptive Parameter Generation Network for Click-Through Rate Prediction

**Bencheng Yan**[*], **Pengjie Wang**[*], **Kai Zhang, Feng Li, Hongbo Deng, Jian Xu, Bo Zheng** [†]

Alibaba Group

China

{bencheng.ybc,pengjie.wpj,victorlanger.zk,
adam.lf,dhb167148,xiyu.xj,bozheng}@alibaba-inc.com

## A   Examples for different condition designs

### A.1   Group-wise

**Example 1 (Thousand users with Thousand Models (a.k.a. Personalized Parameters))** *In    a traditional recommendation model, although some personalized signals (e.g., user-item interactions) are introduced, the shared parameters cannot significantly characterize the personality of each user especially for long-tailed users. In our method, with the help of the group-wise strategy, we can simply give the prior knowledge about users to the model parameters to capture personalized patterns and achieve "Thousand users with Thousand Models". Specifically, given the embedding of a user $m$ as $\boldsymbol{u}_i^{\boldsymbol{m}} \in \mathbb{R}^d$ and an involved instance $i$ of this user, we can directly set $\boldsymbol{z}_i = \boldsymbol{u}_i^{\boldsymbol{m}}$. In other words, by the prior knowledge $\boldsymbol{u}_i^{\boldsymbol{m}}$, we explicitly group the instances by users and allow different users to enjoy different parameters.*

### A.2   Mix-wise

**Example 2 (Real-time Personalized Parameters)** *In practice, user interests may be dynamically changed. For a mix-wise strategy, we can add prior knowledge about users' latest behaviors to allow the parameters sensitive to real-time interests. Specifically, given an instance $i$ associated with a user $m$, we denote $\boldsymbol{z}_i^{\boldsymbol{0,m}} \in \mathbb{R}^d$ as the user $m$ embedding and $\boldsymbol{z}_i^{\boldsymbol{1,m}} \in \mathbb{R}^d$ as the latest behaviors embedding of user $m$ [3]. Then these two conditions $\boldsymbol{z}_i^{\boldsymbol{0,m}}, \boldsymbol{z}_i^{\boldsymbol{1,m}}$ are both considered to generate parameters. In this way, the generated parameters are specific for different users and can be adjusted in real-time by different user interests.*

**Example 3 (Thousand instances with Thousand Models (a.k.a. Instance-aware Parameters)**
*We can consider more conditions to allow the model parameters sensitive to the instance level. To achieve this goal, the conditions are required to identify each instance. Specifically, assuming in a recommendation application, for each instance, we can take the embedding of the associated user, item, and context as $\boldsymbol{z}_i^j$.*

---

[*]* These authors contributed equally to this work and are co-first authors.

[†]† Corresponding author

[3]The latest behaviors embedding can be obtained by averaging the embedding of the latest clicked items of this user

# B The detailed experimental setting

## B.1 Datasets

Four real-world datasets are used: (1)**Amazon** [4] is collected from the electronics category on Amazon. There are total 1,292,954 instances, 1,157,633 users. (2) **MovieLens.** [5] is a review dataset and is collected from the MovieLens web site. There are total 1,000,209 instances, 6,040 users. (3) **IJCAI2018 Advertising Algorithm Competition (IAAC)**[6] is a dataset collected from a sponsored search in E-commerce. Each record refers to whether a user purchases the displayed item after clicking this item. There is a total of 478,138 records, 197,694 users, and 10,075 items. (3) **Industrial dataset (IndusData)** is used for industrial evaluation (see Appendix F). Each instance refers to a user who searches a query in this platform, and the platform returns an item to this user. The label is defined as whether the user clicks this item. There are a total of 4 billion instances, and 100 million users. The statistics of the data sets are summarized in Table 1.

Table 1: The statistic of datasets.

|  | #Data | #User ID | #Item ID |
|---|---|---|---|
| Amazon | 1,292,954 | 1,157,633 | 9,560 |
| MovieLen | 1,000,209 | 6,040 | 3,706 |
| IAAC | 478,138 | 197,694 | 10,075 |
| IndusData | 4 billion | 100 million | 80 million |

## B.2 Baselines.

Here, we compare our method with two kinds of methods

**Existing CTR prediction methods:** To show the effectiveness of the proposed APG, we apply it to various existing deep CTR models (1) WDL[1] adopts wide and deep parts to memorize and generalize patterns of instances. (2) PNN[6] explicitly introduces product operation to explore the interactions of categorical data in multiple fields. (3) FIBINET[3] designs a squeeze-excitation network to dynamically learn the feature importance and use a bilinear-interaction layer to learn the interactions among features. (4) DIFM[4] brings the idea of the transformer and learns vector-wise and bit-wise interactions among features. (5) DeepFM[2] takes the linear part of WDL with an FM network to better represent low-order features. (6) DCN[9] learns low-order and high-order features simultaneously and needs low computation cost. (7) AutoInt[8] learns the feature interactions automatically vim self-attention neural networks.

**Coarse-grained parameter allocating methods:** We also try to compare the proposed APG with the coarse-grained parameter allocating methods: (1) Multi-task learning: MMoE [5] keeps multiple parameters by adopting multiple network branches for different tasks; (2) Multi-domain learning: Star [7] allocates multiple parameters for different scenarios.

## B.3 Training Details

The embedding dimension is set 32 for all methods. The number and units of hidden layers are set $\{256, 128, 64\}$ for all methods by default. Other hyper-parameters of different methods are set by the suggestion from original papers. The backbone of the deep CTR models is set as WDL by default. For APG, we set the self-wise strategy as the default condition strategy. $\mathcal{G}$ is implemented as an MLP with a single layer by default. We set the hyper-parameters $K \in \{2, 4, 6, 8\}$ and $P \in \{32, 64, 128, 256, 512\}$ and perform grid search over $K$ and $P$. We include the results for different values of $K$ and $P$ in Appendix E. We use the Adam optimizer with a learning rate of 0.005 for all methods. The batch size is 1024 for all datasets. Each dataset is randomly split into 80% train, 10% validation, and 10% test sets. For the public datasets, methods are trained on a single V100S

---

[4]https://www.amazon.com/
[5]https://grouplens.org/datasets/movielens/
[6]https://tianchi.aliyun.com/competition/entrance/231647/introduction

Table 2: The AUC (%) results of APG with different kinds of condition strategy, including Group-wise, Mix-wise, and Self-wise. Note Base refers to the results of the method without APG. U, I, and C refers to the embedding of users, items, and contexts respectively.

| Strategy | $z_i$ | MovieLens | Amazon | IAAC |
|---|---|---|---|---|
| Base | | 79.21 | 69.15 | 65.17 |
| Group-wise | U | 79.61 | 69.28 | 65.80 |
| | I | 79.58 | 69.35 | 65.76 |
| Mix-wise | U,I,C | 79.45 | 69.31 | 65.90 |
| Self-wise | $x_i$ | **79.73** | **69.43** | **65.94** |

GPU. For the industrial dataset, methods are trained in an internal cluster equipped with V100S GPU and SkyLake CPU. We run all experiments multiple times with different random seeds and report the average results.

## C  Evaluation of Condition Design

In this section, we analyze the influence of the condition design. Specifically, we evaluate different condition strategies on CTR prediction tasks and report the AUC results. We take WDL as the backbone of APG. For the mix-wise strategy, the input aggregation with attention function is used and the effect of the aggregation method of the mix-wise strategy is detailedly analyzed in Appendix D. Besides we also provide the results of WDL (defined as Base) for comparison.

**The effect of different condition strategies.** We first analyze the effect of different condition strategies, including the group-wise, the mix-wise, and the self-wise strategies. From Table 2, we can find that (1) Compared with Base, all of these condition strategies can obtain better performance. It indicates the effectiveness of APG which can dynamically generate the parameters for better pattern learning; (2) The self-wise strategy achieves the best performance among other strategies in all cases. One of the possible reasons is that the prior knowledge of the self-wise strategy is directly from the hidden layers' input which is a more immediate signal for the current layer and may lead to better parameter generation.

**The effect of different prior knowledge in the same strategy.** We also evaluate the effect of different prior knowledge in the same strategy. Specifically, considering the group-wise strategy, we compare the performance between the prior knowledge about the user embedding and the item embedding. The results are reported in Table 2. We can conclude that (1) Different prior knowledge in the same strategy can get competitive performance; (2) Designing a proper prior knowledge may achieve better performance. For example, for group-wise strategy, taking the item embedding as the prior knowledge obtains higher AUC on Amazon. While the prior knowledge about the user embedding performs better on IAAC and MovieLens.

## D  Evaluation of the Different Aggregation Functions

In this section, we conduct experiments to evaluate the performance when using different aggregation functions for the mix-wise strategy. All of the cases use the embedding of users, items, and the context as the condition. The results are reported in Table 3. We can observe that (1) Compared with other functions, the Attention function achieves the best performance in most cases due to the high representation power of the attention function; (2) Compared with Output Aggregation, Input Aggregation performers better. One of the possible reasons is that the relations among different conditions can be implicitly modeled through $\mathcal{G}$ while output aggregation simply summarizes different specific parameters.

## E  Evaluation of Hyper-parameters

In this section, we conduct experiments to analyze the effect of the hyper-parameters including $P$, $K$, and the number of MLP layers in APG.

Table 3: The results of using different aggratetion function for the mix-wise strategy.

| AUC (%) | Function | MovieLens | Amazon | IAAC |
|---|---|---|---|---|
| Base | | 79.21 | 69.15 | 65.17 |
| Input Aggregation | Mean | 79.33 | 69.28 | 65.44 |
| | Concat | 79.38 | **69.36** | 65.73 |
| | Attention | **79.45** | 69.31 | **65.90** |
| Output Aggregation | Mean | 79.30 | 69.34 | 65.44 |
| | Concat | 79.33 | **69.36** | 65.57 |
| | Attention | 79.41 | 69.25 | 65.70 |

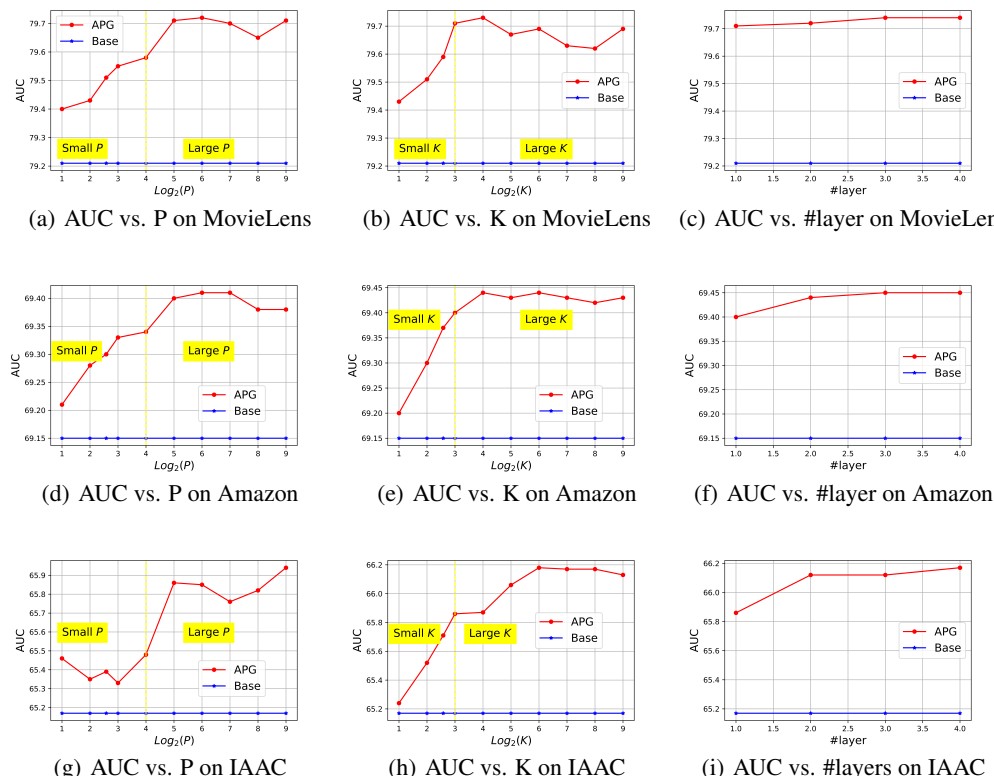

(a) AUC vs. P on MovieLens    (b) AUC vs. K on MovieLens    (c) AUC vs. #layer on MovieLens

(d) AUC vs. P on Amazon    (e) AUC vs. K on Amazon    (f) AUC vs. #layer on Amazon

(g) AUC vs. P on IAAC    (h) AUC vs. K on IAAC    (i) AUC vs. #layers on IAAC

Figure 1: Evaluation of the hyper-parameters.

## E.1 The effect of different $P$

The hyper-parameter $P$ is introduced to add more shared parameters. Thus $P$ is required to be much larger than $K$ (i.e., $P \gg K$). Here we keep $K = 8$ for all cases, and set $P$ to $\{32, 64, 128, 256, 512\}$ respectively and evaluate the performance of APG. The results are plotted in the right part (i.e., Large $P$) in Figure 1 (a)(d)(g). Note we also provide the results of the Base for comparison. We can find that setting different large values of $P$ can give a similar better performance compared with Base. It indicates that when adding more parameters by setting a large $P$, the model can be stably improved.

**Does a small $P$ works?** We further manually set $P$ as a small value (e.g., $\{2, 4, 6, 8, 16\}$). From the left part (i.e., Small $P$) in Figure 1 (a)(d)(g), we can find, that although APG still performs better than Base, there exists performance gap between Large $P$ and Small $P$. It shows the importance of introducing sufficient shared parameters by over parameterization.

Table 4: The results of the severing efficiency. v1 refers to the basic model of APG.

| | Base | APG | v1 |
|---|---|---|---|
| RT(ms) | 14.5 | 14.8 | 57.1 |
| PVR(%) | 0 | -0.01 | -33.2 |
| Memory (M) | 138.03 | 16.14 | 4812.7 |

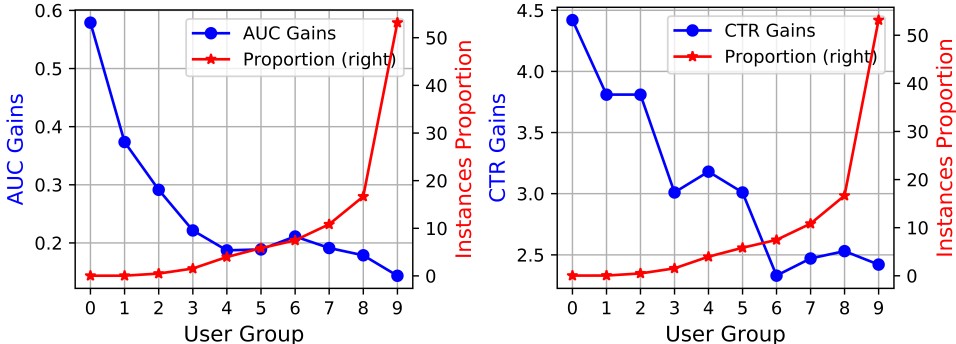

Figure 2: The AUC (**Left**) and CTR (**Right**) gains in different user groups.

## E.2 The effect of different $K$

In this part, we evaluate the effect of different $K$. Specifically, we set $P = 32$ and the hidden layers of the deep CTR model as $\{256, 128, 64\}$ for all cases. Since $K$ is required to be much smaller than $min(N, M)$, we firstly set $K$ to $\{2, 4, 6, 8\}$ respectively. The results are reported in the left part (i.e., Small $K$) in Figure 1 (b)(e)(h). Note we also provide the results of the Base for comparison. We can find that (1) APG performs better than Base in all cases which shows the effectiveness of APG; (2) With the increasing of $K$, APG can also achieve better performance. It indicates that it is important to give sufficient specific parameters to characterize the custom patterns of different instances.

**Does a large $K$ helps?** Similar to the purpose of over parameterization, ignoring the heavy cost of storage and computation, we can also set $K$ to a large value (e.g., $\{16, 32, 64, 128, 256, 512\}$) to see whether the model can obtain further improvements. The results are presented in the right part (i.e., Large $K$) in Figure 1 (b)(e)(h). Some observations are summarized as follows: (1) When we set a large value of $K$, compared with the small one, APG can further improve the model performance in most cases. It indicates increasing $K$ to a large value does help to model the custom patterns; (2) When $K$ is set to extremely large values (e.g., 256 or 512), APG only achieves similar performance with the cases where $K \in \{16, 32, 64, 128\}$; It shows enlarge the specific parameters does not always give a positive contribution and it is wiser to set a suitable value of $K$.

## E.3 The effect of the different number of layers

Here, we increase the number of the MLP layers in APG to evaluate the performance. We also report the results of Base for comparison. The results are depicted in Figure 1 (c)(f)(i). Some observations are summarized as follows (1) Compared with Base, APG with a different number of layers can always achieve significant improvement; (2) Compared with the performance of a single layer, increasing the number of layers can make a slight improvement.

## F  Performance in Industrial Sponsored Search System

Here, we show the performance of APG on industrial applications. Specifically, we firstly train APG with the industrial dataset and then develop it in the sponsored search system. Since December 2021, APG is developed and served as the main traffic of our system.

**Overall Gains.** Compared with the online model, it achieves 0.2% gains in AUC. During the online A/B test, we observe a 3% CTR gain and 1% RPM (Revenue Per Mile) gain respectively. Note this is a significant improvement in the industrial sponsored search system.

**Severing Efficiency.** We further evaluate the severing efficiency of APG. The average Response Time (RT) and Page View Rate (PVR) of the model inference online are evaluated when a user search queries. We also present the memory cost of the hidden layers in each model. Not PVR is influenced by the request timeout. The results are reported in Table 4. Considering the memory efficiency, the memory cost of APG is $8\times$ smaller than Base. Considering the time efficiency, APG does not achieve much improvement and has similar RT and PVR compared to Base. The reason is that, in online serving, the inference time may not have a direct positive relation with GFlops (or theoretical complexity) since the calculation is not always the bottleneck due to the powerful distributed environment and other factors (e.g., I/O, cpu-gpu communication and etc.) also have a great impact on the inference time. In addition, compared with v1, APG achieves a significant improvement in time cost and memory usage, which demonstrates the efficiency of the proposed extensions in APG.

**The effect on low-frequency users.** As described in Introduction Section, adding the specific parameters can capture the custom patterns for different instances, especially for the long-tailed instances, since without the specific parameters the model may be easily dominated by the hot instances. Thus, we detailedly analyze the impact on the different instances when considering the specific parameters. Specifically, we divide the users into 10 groups by their frequency. The frequency is increased from group 0 to group 9 and the number of users is set the same in different groups. Then we evaluate the improvement (including AUC and CTR) of different groups respectively. The results are reported in Figure 2. We can find that (1) Since group 9 refers to the users with the highest frequency, although it has only 10% users, this group produces more than 50% instances (see the red line in Figure 2.); (2) The specific parameters contribute more for low-frequency users since it achieves more gains of AUC and CTR in low-frequency users (e.g., group 0). It demonstrates that specific parameters do allow low-frequency instances to better represent their features, leading to better performance.