# OpenReview forum: "APG: Adaptive Parameter Generation Network for Click-Through Rate Prediction"
_NeurIPS.cc/2022/Conference — NeurIPS 2022 Accept_

### Official Review · Reviewer_gHsX · 2022-06-30

**Rating:** 4
**Confidence:** 4
**Soundness:** 2 fair
**Presentation:** 3 good
**Contribution:** 2 fair

**Summary:**

This paper aims to improve an important application -- CTR prediction.

The authors claim that a static way of parameterization is suboptimal, where the weight of CTR network is shared by all the instances.
Instead, they prefer a dynamic way of parameterization, where a personalized weight is generated for each instance.

Specifically, they apply a hypernetwork to generate the weight for each instance. Moreover, they apply techniques like low-rank parameterization, Decomposed Feed-forwarding, Parameter sharing and Over Parameterization to improve the efficiency and effectiveness of their method.

They present the experimental results on both public datasets and online real-world search system.

**Questions:**

May I ask if authors could read more related papers of hypernetworks and put this work into that context and re-evaluate the novelty?

Is this one the first paper that introduce hypernetworks into CTR prediction?

The current version seems to be a better fit to some application-toward conferences like SIGIR, KDD, considering this paper neither provide deep and theoretical analysis about why same and shared parameters for all instances is suboptimal nor present original idea of hypernetworks.

It is no doubt that this paper is a good application paper that apply hypernetworks for CTR prediction.

**Limitations:**

This paper has no potential negative societal impact.

**Strengths And Weaknesses:**

Strengths:

(1) The paper is well-organized. The motivations for using low-rank parameterization, Decomposed Feed-forwarding, Parameter sharing and Over Parameterization are presented in a very clear way. The figures are very clear and beautiful.

(2) The experiments are comprehensive, including public datasets and online real-world search engine.

Weakness:

(1) The novelty is limited. The core idea of this paper is using a shared hypernetwork to generate weight for each instance or group, which was originally proposed in "Hypernetworks, David Ha, Andrew Dai, and Quoc V Le. 2016." and has been widely applied like in "Parameter-efficient Multi-task Fine-tuning for Transformers via Shared Hypernetworks, ACL 2021" The techniques like low-rank parameterization and Parameter sharing are also not new, which could be found in "HyperGrid Transformers: Towards A Single Model for Multiple Tasks, ICLR 2021."

(2) Related work is missing. Hypernetwork should be an important piece of the related work in the paper but authors seem to completely unaware of the existing of this piece.

(3) The improvement is limited on public datasets. In Table 2, it seems to me that the improvement of APG is not that large. For example, the average gain of APG is only 0.24 in terms of AUC on MovieLens dataset. In particular, the SOTA CTR architecture DIFM has only been improved 0.1, 0.06 and 0.66 on the three datasets, without significant test, it is really hard to be convinced that this method can substantially improve the CTR performance.

---

> ### Author Response · Authors · 2022-08-02
> **Response to Reviewer gHsX (2/2 Part)**
>
> **4、[First hypernetwork paper in CTR prediction? (Q2)]**
>
> Yes. As far as we know, it is the first work to adopt parameter generation for different instances in CTR prediction. We also would like to note that the adoption is not a trivial task. Some special challenges such as computational efficiency and prediction effectiveness need to be addressed to make APG work and successfully serve online in real CTR prediction applications.
>
>
> **5、[Application or Theory or ? (Q3 and Q4)]**
>
> We appreciate your positive comments "It is no doubt that this paper is a good application paper". However, we have some concerns with the concepts of "application papers" and "theory conferences". More details:
>
> * _Theory conference._ NeurIPS is a comprehensive conference that includes various topics, e.g., application, theory, and so on (see Call For Papers in NeurIPS 2022). We believe it is because of such diversity that makes this conference flourish.
> * _Application paper._ It is true that our work focuses on the practical problem of CTR prediction. But it is not a simple application of hypernetworks. On the one hand, a series of novel techniques and architectures are proposed. Novel ideas to address the efficiency and effectiveness problems in CTR prediction are presented in this paper. On the other hand, we also give a theoretical analysis of the complexity of the proposed method.
>
>
> **6、[The improvement is limited on public datasets (W3)]**
>
> The significant improvement of our method on public datasets can be summarized as follows:
>
> * It is a universal module that can be easily applied in most existing deep CTR models and improve the performance in all cases (see Table 2).
> * It is true that in some cases the improvement seems not that large (as mentioned in W3). On the one hand, 0.1% absolute AUC gain is regarded as significant for the CTR prediction task [4,5,6]. On the other hand, DIFM can achieve SOTA performance due to high-consuming-computation operations (see DIFM paper). It is more meaningful where a model (e.g., WDL+APG) with low-computational cost can achieve competitive or even better performance than SOTA DIFM (see Table 2).
> * The significance of APG is not only reflected in the effectiveness (Section 4.2) but also the efficiency (see Section 4.4)
>
>
> [1] Dynamic Filter Networks. NIPS 2016
>
> [2] Hypernetwork. ICLR 2017
>
> [3] HyperGrid Transformers: Towards A Single Model for Multiple Tasks. ICLR 2021
>
> [4] Deep interest network for click-through rate prediction. KDD 2018.
>
> [5] Autoint: Automatic feature interaction learning via self-attentive neural networks. CIKM 2019
>
> [6] Wide & deep learning for recommender systems. RecSys workshop 2016.

---

> > ### Comment · Reviewer_gHsX · 2022-08-05
> > **feedback**
> >
> > I really appreciate your responses, very powerful and touching.
> >
> > But honestly, I don't really learn too much new ideas from this version -- parameters generated by hypernetworks (Hypernetwork. ICLR 2017), parameters sharing (ALBERT: A Lite BERT for Self-supervised Learning of Language Representations, ICLR 2020), low-rank parameterization (HyperGrid ICLR 2021) are all existing methods.
> >
> > Or either you don't have many deep insights about why instance-based learning is better than group-based learning or whole date-based learning, only an intuitive statement "Ideally, besides modeling the common pattern, the parameters should be more adaptive and can be
> > 49 dynamically changed for different instances to capture custom patterns at the same time."
> >
> > Like I mentioned, if this is KDD or Recsys, I will vote for accept without doubt, since you successfully adopt these existing methods to this new domain -- CTR.
> >
> > Plus, I am not picky guy, I give 6 score to the other two papers under my review.

---

> > > ### Author Response · Authors · 2022-08-08
> > > **Response to the Feedback of Reviewer gHsX (3/3 Part)**
> > >
> > > In a neural network with the same or shared parameter, the mapping $\mathcal{A}^y$ is fixed for all instances. Then, such a network can be able to distinguish between features in a subspace of dimension  $n^{2^L}$. While for APG, the mapping $\mathcal{A}^y$  depends on data which enhances the model capacity. Next, we will show the expression power of APG can be bigger than  $n^{2^L}$.
> > >
> > > Specifically, for APG, the difference of $h_y(x_1,...,x_N) - h_y(\hat{x}_1,...,\hat{x}_N)$ is:
> > >
> > > \begin{equation}
> > > h_y(x_1,...,x_N) - h_y(\hat{x}_1,...,\hat{x}_N)
> > > \end{equation}
> > >
> > > \begin{equation}
> > > =\sum_{d_1,...,d_N=1}^{M} \mathcal{A}^y_{d_1,...,d_N} \prod_{i=1}^{N}f_{d_i}(x_i) - \sum_{d_1,...,d_N=1}^{M} \mathcal{\hat{A}}^y_{d_1,...,d_N} \prod_{i=1}^{N}f_{d_i}(\hat{x}_i)
> > > \end{equation}
> > >
> > > \begin{equation}
> > > =\sum_{d_1,...,d_N=1}^{M} \mathcal{A}^y_{d_1,...,d_N} \left( \prod_{i=1}^{N}f_{d_i}(x_i)-\prod_{i=1}^{N}f_{d_i}(\hat{x}_i) \right) ~~~ (2)
> > > \end{equation}
> > >
> > > \begin{equation}
> > > +\sum_{d_1,...,d_N=1}^{M} \left(\mathcal{A}^y_{d_1,...,d_N} - \mathcal{\hat{A}}^y_{d_1,...,d_N} \right) \prod_{i=1}^{N}f_{d_i}(\hat{x}_i) ~~~ (3)
> > > \end{equation}
> > >
> > > If term (2) and term (3) above are both nonzero, having term (2) exactly equal to the negative of term (3) results in zero measure over the space of network parameters, since the parameter generation network is independent from the deep ctr models. Then we can simply consider the cases where term (2) is zero (note it refers to the indistinguishable case in above Eq 1 of the base method) and discuss whether term (3)  is also zero. Here we take the group-wise strategy as an example, since other strategies can be taken as special cases of the group-wise strategy by setting a different number of instances in each group. We assume all instances are dividing into $T$ groups (i.e., $\{G_1,G_2,...,G_T\}$). Then the probability of choosing exactly the same generated parameters for different instances (i.e., $\mathcal{A}^y_{d_1,...,d_N} =  \mathcal{\hat{A}}^y_{d_1,...,d_N}$) is:
> > >
> > > \begin{equation}
> > > \frac{|G_1|}{\sum_{i}|G_i|} \times \frac{|G_1|-1}{\sum_{i}|G_i|-1}+\frac{|G_2|}{\sum_{i}|G_i|} \times \frac{|G_2|-1}{\sum_{i}|G_i|-1}+...+\frac{|G_T|}{\sum_{i}|G_i|} \times \frac{|G_T|-1}{\sum_{i}|G_i|-1}
> > > \end{equation}
> > >
> > > \begin{equation}
> > > =\frac{\sum_{i}(|G_i|(|G_i|-1))}{\sum_{i}|G_i|(\sum_{i}|G_i|-1)}
> > > \end{equation}
> > >
> > > \begin{equation}
> > > =\frac{\sum_{i}(|G_i|(|G_i|-1))}{D(D-1)}
> > > \end{equation}
> > >
> > > where $D=\sum_{i}|G_i|$ refers the number of all instances.
> > > Therefore, with a positivity of the representation functions, the probability of  $h_y(x_1,...,x_N) - h_y(\hat{x}_1,...,\hat{x}_N) \neq 0 $ is
> > >
> > > $1-\frac{\sum_{i}(|G_i|(|G_i|-1))}{D(D-1)} $.
> > >
> > > It means that there is a $1-\frac{\sum_{i}(|G_i|(|G_i|-1))}{D(D-1)} $ probability that the expressive power of APG is bigger than $n^{2^L}$ (the base). Furthermore, from $\frac{\sum_{i}(|G_i|(|G_i|-1))}{D(D-1)}$, we can find that a more sparse group division can lead to a lower value of  $\frac{\sum_{i}(|G_i|(|G_i|-1))}{D(D-1)}$ and result in a higher expressive power. Thus, the self-wise strategy is recommended since the value in each dimension of input $z_i$ in Eq 1 of our paper can be different from different instances.
> > >
> > >
> > > **5、[At Last]**
> > >
> > > We really appreciate your valuable and high-quality reviews, and hope to have addressed your concerns. At last, considering the pioneer and valuable work in CTR prediction, the novel techniques, and the theoretical analysis (in the above point 4), we sincerely hope you could re-evaluate our paper.
> > >
> > >
> > > [1]  Hypernetwork. ICLR 2017
> > >
> > > [2]  HyperGrid Transformers: Towards A Single Model for Multiple Tasks. ICLR 2021
> > >
> > > [3] ALBERT: A Lite BERT for Self-supervised Learning of Language Representations, ICLR 2020
> > >
> > > [4] How Large a Vocabulary Does Text Classification Need? A Variational Approach to Vocabulary Selection, NAACL 2019
> > >
> > > [5]  On the expressive power of deep learning: A tensor analysis. In Conference on Learning Theory, 2016.

---

> > > ### Author Response · Authors · 2022-08-08
> > > **Response to the Feedback of Reviewer gHsX (2/3 Part)**
> > >
> > >
> > > **4、[Deep insight about APG]**
> > >
> > > Thanks for your advice on giving a deep insight about APG. We agree conducting the theoretical analysis can better support and understand APG, and make this work stronger.
> > >
> > > Here, we give a theoretical analysis of the expressive power of APG and the guidelines of condition design.
> > > We will add this part to the revised manuscript.
> > >
> > > Inspired by Cohen et al. [5], which establishes a tensor analysis approach to prove the expressive power of a deep neural network increases super-exponentially with the network depth, based on the network width, we also characterize the expressive power of APG in this tensor perspective.
> > >
> > > To begin,  an instance is defined as a collection of vectors $(x_1,...,x_N)$, where $x_i \in \mathbb{R}^s$ refers to a feature vector of this instance. Then we can represent these different features by a (positive) representation function:
> > >
> > > \begin{align}
> > > f_{d_i}(x_i)
> > > \end{align}
> > >
> > > where $d_i \in [1,2,..,M]$ and $M$ is the number of different feature representation. Then, for a classification task, we take a deep neural network as a mapping from an instance to a cost function over label $y$. Following Cohen et al. [5], such a mapping can be represented by a tensor $\mathcal{A^y}$ operated on the combination of the representation functions:
> > >
> > > \begin{align}
> > > h_y(x_1,...,x_N)=\sum_{d_1,...,d_N=1}^{M} \mathcal{A}^y_{d_1,...,d_N} \prod_{i=1}^{N}f_{d_i}(x_i)
> > > \end{align}
> > >
> > > Then the expressive power of this network is defined as the ability to construct labeling to differentiate input values. More precisely, to be able to distinguish data instances $x$ from $\hat{x}$ ($x \neq \hat{x}$),   $h_y(x_1,...,x_N) - h_y(\hat{x}_1,...,\hat{x}_N)$ is required to be nonzero, i.e.,
> > >
> > > \begin{align}
> > > \sum_{d_1,...,d_N=1}^{M} \mathcal{A}^y_{d_1,...,d_N} \left( \prod_{i=1}^{N}f_{d_i}(x_i)-\prod_{i=1}^{N}f_{d_i}(\hat{x}_i) \right) \neq 0 ~~~ (1)
> > > \end{align}
> > >
> > > It can  be seen that the inequality is held when the difference $\prod_{i=1}^{N}f_{d_i}(x_i)-\prod_{i=1}^Nf_{d_i}(\hat{x_i})$ is not in the null space of $\mathcal{A}^y_{d_1,...,d_N}$. Hence, the expressive power is equivalent to the rank of the tensor $\mathcal{A}^y$. Then the rank of tensor $\mathcal{A}^y$ scales as $n^{2^L}$ with measure 1 over the space of all possible network parameters ($n$ is the network width and $L$ is the network depth) [5].

---

> > > ### Author Response · Authors · 2022-08-08
> > > **Response to the Feedback of Reviewer gHsX (1/3 Part)**
> > >
> > > Thanks for your valuable comments. Please allow us to further address your concerns below.
> > >
> > > **1、[Novelty]**
> > >
> > > As mentioned in points 2 and 3 in the previous response, we have shown:
> > > * For hypernetworks [1], the pioneer work in CTR prediction can be recognized in our paper.
> > > * For HyperGrid [2], although similar techniques  (e.g., low-rank parameterization) are adopted in HyperGrid  and ours,  the novelty in motivations behind these techniques are more important and could be appreciated
> > > * Apart from some similar techniques (mentioned in reviews), we also design a number of novel techniques in this paper (e.g., decomposed feed-forwarding, over-parameterization, and condition strategies).
> > >
> > > Besides, thanks for pointing out a  new related paper ALBERT [3] mentioned in this feedback. However, the parameter sharing used in [3] is a different concept from our paper. For ALBERT, parameter sharing refers to sharing parameters across different layers. For APG, parameter sharing refers to sharing parameters across different instances in the same layer. Furthermore, ALBERT is also not in the concept of hypernetworks and it focuses on the model reduction of Bert.
> > >
> > > **2、[Valuable work]**
> > >
> > > Apart from the novelty of our paper (mentioned in the above point 1), we also want to highlight that it is a  valuable work:
> > >
> > > * New direction. In the current stage, most of the existing CTR-related works focus on architecture design and ignore the improvement of the model parameters $\Theta$. We believe the idea of APG can bring a new direction and encourage many follow-up works.
> > >
> > > *  Universal module.  The proposed APG is a universal module that can be easily applied to most existing deep CTR models and improve the performance in all cases (see Table 2).
> > >
> > > *  Real industrial application.  We have successfully deployed APG in an industrial sponsored search system and achieved significant gains.
> > >
> > > All in all,  we show the novelty of the ideas and techniques in this work in the above point 1. At the same time, we hope the valuable parts and further influence of this work could be recognized, which could also play an important role  in evaluating whether a paper is good or not.
> > >
> > > **3、[Not a simple application]**
> > >
> > > We would like to re-emphasize that APG is not a simple application. There naturally exist gaps between CV/NLP and CTR prediction, which leads to great challenges. More details:
> > >
> > > * Time sensitive.  Different from CV/NLP, for the practical problem of CTR prediction, users are sensitive to the response time when they are searching in a recommendation system, and a slight increase in inference time might not be tolerated.  More seriously, there has a heavy dense matrix product in CTR prediction models, which further aggravates the efficiency problem (see Section 1 in this paper). Hence, settling such a challenge significantly supports the novelty and the valuable idea of this paper.
> > > *  Big vocabulary size.  Compared with a limited vocabulary size used in NLP (e.g., ten thousand unique words [4]),  for CTR prediction,  the data is constructed by a large number of users and items (e.g., 100 million users and 80 million items in this paper). A huge vocabulary size gap drives us to carefully model the correlation and the diversity among different instances. Hence, two kinds of parameters (including shared and specific parameters) are proposed and various condition strategies are designed.
> > >
> > > Here, we really hope reviewer gHsX could think twice about the difference between CV/NLP and CTR prediction rather than  directly noting “application paper”.  It's these differences that make our work valuable and inspire the novel techniques in this paper.

---

> ### Author Response · Authors · 2022-08-02
> **Response to Reviewer gHsX (1/2 Part)**
>
> Thanks for your valuable comments. Please allow us to address your concerns below.
>
> **1、[Lack of related works of hypernetwork (W1、W2 and Q1)]**
>
> The idea of generated weights for different instances was originally proposed in DFN [1] which inspired us and has been cited in our paper. Hypernetwork [2] also claims its work is an extension of DFN (see Section 2 in [2]). Considering this, we cited DFN in the current version. Hypernetwork is also an important work that has a great impact on related areas and we will include it and other related work thoroughly in the revised manuscript. Thanks for pointing this out.
>
> **2、[Not new idea of low-rank parameterization and Parameter sharing (W1)]**
>
> Thanks for pointing out a related paper HyperGrid [3]. We agree that HyperGrid seems to adopt similar techniques including low-rank parameterization and parameter sharing. But we would like to note that the motivation is not the same.
>
> * For HyperGrid, the key problem is the task conflict in a single model. To address this problem, the authors introduce local-global parameters which lead to a decomposed implementation and the low parameter cost introduced by low-rank parameterization is the result rather than the motivation. Besides, HyperGrid cares more about the parameter cost difference between multiple models and a single model rather than the cost difference between w/ and w/o low-rank parameterization.
> * For ours, the key problem is the efficiency problem in CTR prediction. Thus low-rank parameterization is proposed to address this problem. A direct justification is that for the sake of efficiency, only S_i is used as the specific parameter. This is also different from HyperGrid.
>
> Although there may be some similar techniques used in different works at the first glance, we appreciate that the novelty in motivations behind these works can be recognized.
>
> **3、[Novelty (W1 and Q1)]**
>
> We also want to re-highlight some of the novel aspects of this paper and will make the presentation of the novelty more explicit in the revised manuscript.
>
> * It is true that the idea of generating parameters has been widely studied in CV and NLP. But they do not diminish the novelty of our work. We believe such a direction in CTR prediction can boost and encourage many follow-up works, and create chances of cooperation among CV, NLP, and CTR prediction. Furthermore, we also would like to share some cases: the main contribution of swim transformer (the ICCV 2021 best paper) is successfully introducing transformer from NLP to CV.
> * It is not a simple attempt or application where both the efficiency and effectiveness are key in CTR predictions. There is no trivial solution to address these challenges. Therefore, the novel method APG is proposed. More details:
> 	* Apart from low-rank parameterization and parameter sharing mentioned in the above point 2, we want to emphasize the novelty of the proposed decomposed feed-forwarding. Actually, it is the key to addressing the computation efficiency problem and allowing APG to be efficiently serving online.
> 	* We also want to highlight the novelty of over-parameterization which enriches the model capacity without any additional memory and time cost during inference.
> * We propose novel condition strategies which are seldom discussed in most related works in CV and NLP.
>
> All in all, considering the pioneer work in CTR prediction and the novel techniques proposed in this paper (e.g., decomposed feed-forwarding, over-parameterization, and condition strategies), we sincerely hope reviewer gHsX could re-judge the novelty of our paper.

---

### Official Review · Reviewer_Zx1w · 2022-07-12

**Rating:** 6
**Confidence:** 3
**Soundness:** 3 good
**Presentation:** 3 good
**Contribution:** 3 good

**Summary:**

This paper proposes to learn input-aware parameters in deep CTR models to boost its representation power. A clear method iteration trace is presented with clear design and solid analysis. The final version archives both efficiency gain and accuracy improvement. Most impressively, the method is tested in industrial production system.

**Questions:**

The novelty of such fine-grained input-aware parameter allocation methods. The related work section states that this is almost first work to apply such heuristic, right?

Do we have benchmark on the effectiveness of various condition designs?

The gain from v4 to v5 is huge, why an over parameterization trick so powerful? If parameter sharing is beneficial, base model should not be such poor?

**Limitations:**

The method iteration are surprisingly positive which may imply some deep insights. Let's say
- base -> v1 parameter personalization improves performance
- v1 -> v2 low rank parametrization doesn't change performance too much.
- v2 -> v4 appropriate parameter sharing improves performance
- v4 -> v5 over-parameterization (more sharing parameters) largely improves performance

It seems model favors heavy parameter sharing plus certain parameter personalization. More effort is needed to study the root factor and a mode direct approach based on it.

**Strengths And Weaknesses:**

Strengths
The paper is well written and quite readable. Methods' iteration records are plausible.

Weaknesses
Current baseline on parameter allocation is too weak. This is not necessarily a weakness since advanced baseline to learn fine-grained parameters for model parameters may not be existed. I am not an expert of related literatures.

---

> ### Author Response · Authors · 2022-08-02
> **Response to Reviewer Zx1w**
>
> Thanks for your valuable comments. Please allow us to address your concerns below.
>
> **1、[Q1:The novelty of such fine-grained input-aware parameter allocation methods. The related work section states that this is almost first work to apply such heuristic, right?]**
>
> Yes. As far as we know, it is the first work to adopt parameter generation for different instances in CTR prediction. We also would like to note that the adoption is not a trivial task. Some special challenges such as computational efficiency and prediction effectiveness need to be addressed to make APG work and successfully serve online in real CTR prediction applications.
>
>
> **2、[Q2:Do we have benchmark on the effectiveness of various condition designs?]**
>
> As mentioned in Appendix C, empirically speaking, the self-wise strategy shows the best performance and it is also easy to apply since it does not need any additional prior knowledge. Of course, sometimes, you may have a special purpose. For example, if you want to build different model parameters for different users, you can consider the group-wise strategy.
>
>
> **3、[Q3:The gain from v4 to v5 is huge, why an over parameterization trick so powerful? If parameter sharing is beneficial, base model should not be such poor?]**
>
> Actually, from Table 4, the gains come majorly from allowing parameters to be sensitive to different instances rather than over-parameterization. Specifically, the gains for different techniques are 0.27(adding specific parameters from base to v1), 0.13(adding shared parameters from v2 to v4), and 0.12 (adding over-parameterization from v4 to v5).
>
> The gains of over-parameterization come from the following parts:
>
> * Without over-parameterization, the amount of shared parameterization is not enough (due to small K) to model the common patterns. Thus, adding over-parameterization can significantly improve the performance. While for the base model, the shared parameters are large enough, and simply increasing the number of parameters may not give much improvement. Besides, from Appendix E.1, we can also find when P is large enough, the gain is convergent.
> * Actually, as mentioned in Section 3.2, over-parameterization can result in an implicit regularization and thus enhance generalization.
>
>
> **4、[More effort is needed on Ablation Study (L1)]**
>
> Thanks for your advice on paying more effort to the ablation study. We will make a more detailed analysis in the revised manuscript. Here, we briefly summarize the key points of the improvement with different versions:
>
> * base -> v1, as described in Section 1, parameter personalization can enrich the expression power of deep CTR models.
> * v1 -> v2, since the weight matrix resides on a low intrinsic dimension [1,2], it encourages a low-performance drop when adopting a low-rank based method. Besides, we also conduct a detailed experiment to analyze the impact of the rank k (see Appendix E.2).
> * v2 -> v4, as mentioned in Section 3.2, parameter sharing can capture common and general information which improves the performance.
> * v4 -> v5, see the above answer (i.e., point 3).
>
> **5、[Weak baseline on parameter allocation.(W1) ]**
>
> As far as we know, currently, almost all related works in CTR prediction focus on coarse-grained parameter allocation. Thus, if a strong baseline needs to be established, a possible perspective is to directly take the basic model (proposed in Section 3.1) as a strong baseline.
>
> [1] Measuring the intrinsic dimension of objective landscapes. arXiv 2018
>
> [2] Intrinsic dimensionality explains the effectiveness of language model fine-tuning. arXiv 2020

---

> > ### Comment · Reviewer_Zx1w · 2022-08-10
> > **Thanks for the response.**
> >
> > Thanks and I don't have further questions.

---

### Official Review · Reviewer_QUZk · 2022-07-12

**Rating:** 7
**Confidence:** 3
**Soundness:** 3 good
**Presentation:** 3 good
**Contribution:** 3 good

**Summary:**

This paper studies the problem of enabling input-aware model parameters which are dynamically generated in order to boost representation power of deep CTR (click-through rate) prediction models. A novel and general method APG (Adaptive Parameter Generation) is proposed, being able to work together with a variety of existing CTR prediction models. By means of employing several techniques such as low-rank parameterization, decomposed feed-forwarding, parameter sharing and over parameterization, APG generates adaptive parameters in an efficient and effective way. Computational complexity is analyzed carefully; empirical results demonstrate APG's efficiency and effectiveness. Notably, APG is claimed to be deployed in an industrial sponsored search system, achieving performance gain in online A/B testing.

**Questions:**

In Line 283 “Since over parameterization does not introduce any cost, it is not considered here”: Why? It seems that over parameterization would introduce more computational cost during training.




**Ethics Review Area:**

["I don’t know"]

**Limitations:**

The writing has to be polished carefully. Several typos are listed as below:

* Line 87: analysis → analyze
* Line 106: is lack → lacks
* Line 107: kinds → kind
* Line 118: General → Generally
* Line 215: detailed analyze → analyze in detail
* Line 242: Since APG → APG
* Line 282: analysis → analyze
* Line 349: detailed analyzed → analyzed in detail

**Strengths And Weaknesses:**

* **Originality**: Dynamic neural networks are widely studied in computer vision and natural language processing, in which model parameters and architectures could be input-aware and thus dynamically generated. It is claimed that this paper is the first to bring the idea about dynamic neural networks into deep CTR prediction models.
* **Clarity**: The paper is well organized and clearly presented. However, typos are widespread throughout the text.
* **Significance**: Computational complexity of the proposed APG is carefully analyzed, and the efficiency is also validated empirically. A wide variety of baselines are employed to demonstrate the generality of APG. It is noteworthy that APG is claimed to be deployed in an industrial sponsored search system, achieving performance gain in online A/B testing.

---

> ### Author Response · Authors · 2022-08-02
> **Response to Reviewer QUZk**
>
> Thanks for your valuable comments. Please allow us to address your concerns below.
>
> **1、[Q1:In Line 283 “Since over parameterization does not introduce any cost, it is not considered here”: Why? It seems that over parameterization would introduce more computational cost during training.]**
>
> We sincerely apologize for our misleading writing in the current version in Table 5. Actually, the time refers to inference time in Table 5, and for CTR prediction we care more about the online inference efficiency. As mentioned in Section 3.2, over-parameterization does not introduce any additional latency or memory cost to inference. It means the time and memory cost is similar to V4. It is true that adding over-parameterization will bring additional cost to training.  Again, we are sorry for the misleading expression and we will fix it  in the revised version.
>
>
> **2、[typos (L1)]**
>
> Thanks for your careful review. We will definitely polish the paper and a thorough proofreading will be conducted.

---

### Official Review · Reviewer_2in4 · 2022-07-15

**Rating:** 6
**Confidence:** 4
**Soundness:** 3 good
**Presentation:** 4 excellent
**Contribution:** 3 good

**Summary:**

The authors of the paper address the problem of click-through rate (CTR) prediction customized to different instances. They propose an adaptive parameter generation (APG) method for generating weight matrices in instance-adapted CTR systems. Generating the entire weight matrix directly is very time and memory consuming. To reduce time and memory complexity, they propose to decompose the weight matrix into a product of three vectors, namely S, U, and V. Only S is dynamically generated to capture different instances of custom patterns, while U and V are shared instances that capture common patterns. Results from three public datasets and real search systems are shown.

**Questions:**

1. In a group-wise conditional design, is there any extra effort required to divide instances into different groups?
2. S is used to capture custom patterns because of its low rank. Have you tried using U and V for this purpose? Did they achieve similar results?
3. V5, over parameterization, appears to be a separate technology. How it connects to APG.
4. What is the APG version reported in Table 2?
5. The training time and memory complexity of v5 is not provided in Table 5. Is it similar to the basic version? What is the trade-off between the efficiency and effectiveness of v5?

**Limitations:**

1. Can provide a comparison between using generated and fixed S in the current v1-v5 version.
2. Can provide comparisons with existing methods of automatically generating weight matrices (e.g., in the AutoML domain).

**Strengths And Weaknesses:**

Strengths:
1. By decomposing the matrix and then decomposing the feedforward, the time and memory complexity of the CTR model with APG is greatly reduced.
2. Improve scalability to large inputs through over-parameterization.
3. The paper is well written.

Weakness:
1. Does not provide a comparison of between generating the S vector and using fixed S vector on the current v1-v5 versions.
2. It does not provide a comparison with existing methods that automatically generate weight matrices (e.g., in the AutoML field).
3. A small typo on line 210.

---

> ### Author Response · Authors · 2022-08-02
> **Response to Reviewer 2in4**
>
> Thanks for your valuable comments. Please allow us to address your concerns below.
>
> **1、[Q1: In a group-wise conditional design, is there any extra effort required to divide instances into different groups?]**
>
> Group-wise conditional design can be flexible. One can simply divide instances into different groups by different item categories, since this kind of information is usually directly available. One can also divide instances into different groups with some clustering methods.
>
>
> **2、[Q2: S is used to capture custom patterns because of its low rank. Have you tried using U and V for this purpose? Did they achieve similar results?]**
>
> Thanks for your advice. Here we conduct additional experiments by using U and V as specific parameters.
> The results are as follows:
>
> |Version            | MovieLens | Amazon | IAAC  | Ave(AUC) | Ave($\Delta$ ) |
> | -------- | -------- | -------- |-------- |-------- |-------- |
> |$U_i(S(V_ix_i))$ |   79.64	 |   69.27    | 65.80 |  71.57       | +0.39 |
>
> Generally speaking, although using both U and V can achieve similar performance, it is costly compared to using only S. The reason is that the generation complexity of the specific parameter is sensitive to N and M when using U and V.
>
>
> **3、[Q3: V5, over parameterization, appears to be a separate technology. How it connects to APG.]**
>
> As mentioned in Section 3.2, although low-rank can significantly reduce the model complexity, the number of the shared parameter is reduced which may hurt the model performance. Thus, over-parameterization is proposed.
>
>
> **4、[Q4: What is the APG version reported in Table 2?]**
>
> It is V5 in Table 2 and we are sorry to make you confused about the version in Table 2. We will clarify it in the revised manuscript
>
>
> **5、[Q5: The training time and memory complexity of v5 is not provided in Table 5. Is it similar to the basic version? What is the trade-off between the efficiency and effectiveness of v5?]**
>
> We sincerely apologize for our misleading writing in the current version in Table 5. Actually, the time refers to inference time in Table 5, and for CTR prediction we care more about the online inference efficiency. As mentioned in Section 3.2, over-parameterization does not introduce any additional latency or memory cost to inference. It means the time and memory cost is similar to V4. It is true that adding over-parameterization will bring additional cost to training. But considering the efficiency during inference, for the CTR prediction tasks, it is willing to adopt over-parameterization.
>
> Again, we are sorry for the misleading expression and we will make it clear in the revised version.
>
> **6、[L1 and W1: Can provide a comparison between using generated and fixed S in the current v1-v5 version.]**
>
> Here we provide additional results of the version using fixed S (i.e.,$U(S(Vx_i))$ ).
>
> |Version | MovieLens | Amazon | IAAC  | Ave(AUC) | Ave($\Delta$ ) |
> | -------- | -------- | -------- |-------- |-------- |-------- |
> |$U(S(Vx_i))$  |   79.09	       |   69.21    | 65.26 |  71.19       | +0.01|
>
> We can see that using fixed S achieves similar performance to v1.
>
>
> **7、[L2 and W2: Can provide comparisons with existing methods of automatically generating weight matrices (e.g., in the AutoML domain).]**
>
> Thank you for your nice idea to consider the comparison with AutoML methods. As far as we know, the methods using AutoML are more likely to be used for automatically selecting operations or architectures like [1,2] rather than weight generation. We will also discuss the relationship with AutoML in the revised manuscript.
>
> **8、[typos (W3)]**
>
> Thanks. We will fix it.
>
> [1] AutoGroup: Automatic feature grouping for modeling explicit high-order feature interactions in CTR prediction, SIGIR 2020
>
> [2] Neural input search for large scale recommendation models. KDD 2020

---

> > ### Comment · Reviewer_2in4 · 2022-08-09
> > **I maintain my original rating.**
> >
> > Thanks to the author for the detailed reply. Most of my concerns have been addressed. I keep my original rating.

---

### Meta-Review · Area_Chair_sjNs · 2022-08-28

**Recommendation:** Accept
**Confidence:** Less certain

**Metareview:**

The paper focuses on the application of click-through rate (CTR), and proposes input-aware model parameters which are dynamically generated in order to boost representation power of deep CTR prediction models. To reduce time and memory complexity, the method decomposes the parameters and dynamically generates only part of the decomposed parameters. Improved results are shown on three public datasets and an A/B testing on an industrial system as claimed. Overall, this is a nice application-focused work that applies the widely studied idea of parameter generation and decomposition onto the new problem of CTR.

**Award:**

No

---

### Decision · Program_Chairs · 2022-09-14

Accept